# Psychometric Evaluation of the Malay Version of the Family Adaptability and Cohesion Evaluation Scale III for Malaysian Adolescents

**DOI:** 10.3390/ijerph19010156

**Published:** 2021-12-24

**Authors:** Chin Wen Cong, Chee-Seng Tan, Hooi San Noew, Shin Ling Wu

**Affiliations:** 1Department of Psychology and Counselling, Faculty of Arts and Social Science, Universiti Tunku Abdul Rahman (UTAR), Kampar 31900, Malaysia; 2Department of Communication, College of Liberal Arts, Wenzhou-Kean University, Wenzhou 325060, China; sun.noew@gmail.com; 3Department of Psychology, School of Medical and Life Sciences, Sunway University, Petaling Jaya 47500, Malaysia; shinling_wu@hotmail.com

**Keywords:** adolescent, Circumplex Model, FACES-III, family functioning, Malaysia, psychometrics

## Abstract

The Family Adaptability and Cohesion Scale III (FACES-III) has been widely used to measure an individual’s family functioning in terms of cohesion and adaptability. In Malaysia, the FACES-III has been translated into the Malay language for the community, but its psychometric properties in this context remain unknown. Thus, the purpose of this research is to examine the psychometric properties of the Malay version of the FACES-III in 852 adolescents attending secondary schools in Kuala Lumpur, Malaysia. Data were randomly split into two halves: the exploration sample and the validation sample. Exploratory factor analysis was conducted on the exploration sample and a two-factor model was discovered after removing nine items that showed low factor loading. Then, confirmatory factor analysis was conducted on the validation sample to compare the one-factor models, two-factor models, and three-factor models. Results showed that the 11-item two-factor model (FACES-III-M-SF) was superior to the other competing models. Both the exploratory and confirmatory factor analyses replicated the two-factor structure of the original version of FACES-III. The reliability of the overall scale was consistently good, but the subscale results were mixed. This suggests that researchers should use the overall score, but not the subscale scores, in analyses.

## 1. Introduction

A family is an ideal place for individuals to develop physically and mentally [1,2]. Indeed, family is critical for adolescents who are experiencing major emotional, cognitive, and social changes [3,4]. Healthy family functioning is a protective factor, whereas poor family functioning is a risk factor for a variety of adolescent mental health problems. For instance, family functioning has been linked with adolescent behavioral and emotional problems [5], depressive symptoms [6], obsessive-compulsive disorder [7], and suicide attempts [8].

According to the Circumplex Model of Marital and Family Systems [9], family functioning is characterized by two dimensions—namely, cohesion and adaptability. Cohesion refers to the emotional bonding between family members, while adaptability is the family’s capacity to adopt leaderships, roles, and rules in response to needs [10]. Based on the Circumplex Model, Olson developed the Family Adaptability and Cohesion Evaluation Scale (FACES-III) to assess family functioning level [11]. The FACES-III consists of two subscales (cohesion and adaptability) and shows an acceptable reliability (Cronbach’s alphas were 0.68, 0.77, and 0.62 for the overall scale, cohesion, and adaptability subscales) in a mixed sample of adolescents and adults [11]. On the other hand, in a sample of adolescents Ide et al. (2010) reported a better internal reliability for the FACES-III, with a Cronbach’s alpha of 0.86 for the entire scale and 0.89 and 0.70 for the cohesion and adaptability subscales, respectively.

The FACES-III also shows good validity across different cultures. For instance, the conceptual two-factor structure of the FACES-III has also been found in the Spanish [12] and Japanese versions [13]. However, some items from the adaptability subscale (e.g., “our family changes its way of handling tasks”) of the Japanese FACES-III were found to load on the cohesion factor [13]. This might be due to the Japanese belief that family members should be cohesive in order to handle many tasks. The cross-loading of items also implies that culture may shape the contents and meaning of family functioning [13]. Therefore, researchers are advised to examine the psychometric qualities of the family functioning measurements before employing them in different cultural contexts and populations.

Olson developed a newer version of the FACES, FACES-IV, to address the limitation of FACES-III in measuring the extreme levels of cohesion and adaptability as proposed in the Circumplex Model [14]. Although the FACES-IV is more comprehensive, it is not as widely employed by researchers as the FACES-III. A possible reason for this may be that the FACES-IV requires payment. Researchers in developing countries (e.g., Malaysia), with limited or no funding, tend to use the FACES-III, which is available free of charge. In fact, FACES-III has been widely used to measure family functioning in Malaysia (e.g., [15,16,17]).

To help participants understand the items, Ng and Sulaiman translated the FACES-III into the Malay language (FACES-III-M) using back translation method, and they reported a Cronbach’s alpha of 0.85 for the overall scale in a sample of adolescents [15]. However, the researchers did not report on the reliability of the two subscales, which raises questions about the reliability of the family functioning dimensions. Moreover, we are not aware of studies on the validity of the FACES-III-M in the Malaysian context. That is, the validity of the FACES-III-M has yet to be established in Malaysia. Therefore, the purpose of the present study is to examine the psychometric properties of the FACES-III-M for Malaysian adolescents to bridge the gap of psychometric evidence in the Malaysian context. In particular, using the data of a past study (i.e., [18]), we examined the factorial structure of the FACES-III-M to clarify whether the conceptual two-factor solution applies to Malaysian adolescents, as well as the suitability of the items for capturing the meaning of family functioning. Besides that, the reliability and validity of the FACES-III-M were also examined to offer insights into its usefulness in the context of Malaysia.

## 2. Materials and Methods

### 2.1. Participants and Procedure

A total of 852 secondary school students (50.6% males) were recruited from three different schools in Kuala Lumpur, Malaysia. Their mean age was 14.8 (SD = 1.23), ranging from 13 to 17 years old. Of the sample, 61.6% were Malays, followed by Chinese (31.1%), Indians (4.3%), and other ethnicity groups (3.0%). Approvals were obtained from the Malaysian Ministry of Education (KPM.600-3/2/3-eras(2408)), the Kuala Lumpur Education Department (JPNWP.900-6/1/7 Jld.21(50)), and the schools’ principals prior to the data collection. Data were collected using a paper-and-pencil questionnaire. Participants were asked to read the study information sheet and sign informed consent forms to be involved in the study. The data collection procedure was reviewed and approved by the Scientific and Ethical Review Committee of Universiti Tunku Abdul Rahman (Ref: U/SERC/19/2019). This study is part of a larger project on adolescents’ suicidal ideation and depression (see [18,19,20]).

### 2.2. Measurement

The Family Adaptability and Cohesion Evaluation Scale—Malay version [15]—consists of two subscales: cohesion and adaptability. The cohesion subscale, with 10 items, assesses the degree of connectedness between family members; whereas, the adaptability subscale, with another 10 items, refers to the extent of flexibility and ability to change in a family. The items were rated using a five-point Likert-type scale ranging from 1 (almost never) to 5 (almost always). The total score was computed, with a higher score indicating a higher level of family functioning. Likewise, higher scores in each subscale also indicate higher levels of functioning in that dimension.

### 2.3. Statistical Analysis

The data were randomly split into two halves: the exploration sample (*n* = 424) and the validation sample (*n* = 428). The JASP software package (version 0.13.1) was used to conduct an exploratory factor analysis (EFA) and confirmatory factor analysis (CFA). After running a parallel analysis, the factorial structure underlying the FACES-III-M was examined using EFA on the exploration sample. Then, a CFA was conducted on the validation sample using maximum likelihood estimation to compare the competing models and identify the best fit model.

The model fit was assessed using indices such as the ratio of chi-square values to degrees of freedom (χ^2^/df), Comparative Fit Index (CFI), Tucker–Lewis Index (TLI), Root Mean Square Error of Approximation (RMSEA), and standardized root mean square residual (SRMR). A good fit model would have χ^2^/df < 3, TLI and CFI > 0.95, RMSEA ≤ 0.05, and SRMR < 0.08 [21,22,23].

The reliability of the FACES-III-M and its subscales was examined using the Cronbach alpha (α) and McDonald omega (ω) coefficients. The internal consistency is supported if the two coefficients are greater than 0.70 respectively. Meanwhile, average variance explained (AVE) and its square root value were used to examine the convergent and discriminant validity of the FACES-III-M, respectively. Specifically, convergent validity is supported if AVE > 0.50, while discriminant validity is evident if the square root of the AVE of a subscale is greater than its association with another subscale [24].

## 3. Results

### 3.1. Exploratory Factor Analysis Results and Discussion

An EFA was conducted to examine the factorial structure of the FACES-III-M in the Malaysian context. Following the recommendation of parallel analysis, a two-factor solution using maximum likelihood and Promax rotation was examined. The sampling adequacy measure reported a Kaiser–Meyer–Olkin (KMO) value of 0.896. Bartlett’s test of sphericity was statistically significant, χ^2^(190) = 2188.56, *p* < 0.001, supporting the appropriateness of factorability. The two-factor solution explained 32.3% of the total variance. However, some items were found to have factor loading below 0.40. Thus, the items with a low factor loading were removed (one per time) and then the EFA was re-run. A total of six items (items 2, 4, 5, 8, 16, and 17) were removed. The factorability of the 14 items was also supported, KMO = 0.861, Bartlett’s test of sphericity χ^2^(91) = 1389.18, *p* < 0.001. Similarly, a two-factor solution was found and all items loaded on the target factor (see Table 1). The first factor (cohesion) with items 1, 3, 7, 9, 11, 13, 15, and 19 accounted for 22.6% of the variance (eigenvalue = 3.165), whereas the second factor (adaptability) with items 6, 10, 12, 14, 18, and 20 explained 11.9% of the variance (eigenvalue = 1.668). Overall, the two-factor model with 14 items accounted for 34.5% of the total variance.

Considering that the two-factor model with 14 items explained less than 50% of the total variance, we further removed three items with a factor loading below 0.50 (i.e., items 7, 10, and 20) and re-ran the EFA. The factorability of the 11 items was supported, with a KMO = 0.856, Bartlett’s test of sphericity χ^2^(55) = 1111.8, *p* < 0.001. Again, a two-factor solution was found and the 11 items were loaded on the corresponding factor (see Table 1). The first factor (cohesion) with items 1, 3, 9, 11, 13, 15, and 19 accounted for 26.6% of the variance (eigenvalue = 2.926), whereas the second factor (adaptability) with items 6, 12, 14, and 18 explained 11.6% of the variance (eigenvalue = 1.276). Overall, the two-factor model with 11 items accounted for 38.2% of the total variance. Pearson’s correlation analysis showed that there was a positive relationship between the two subscales, r(422) = 0.217, *p* < 0.001.

The Cronbach’s alpha coefficient for the 11-item FACES-III-M was 0.778 (ω = 0.772). The cohesion and adaptability subscales also demonstrated a high to acceptable internal reliability, with α = 0.829 and ω = 0.831 for the cohesion subscale and α = 0.641 and ω = 0.643 for the adaptability subscale. 

The convergent validity of the 11-item FACES-III-M was not satisfactory, with AVE values below 0.50 in its subscales (i.e., 0.418 and 0.313 for the cohesion and adaptability subscales, respectively). Although the convergent validity was not supported in the present study, the 11-item FACES-III-M established a discriminant validity (i.e., the square root of the AVE of both subscales was greater than the factor covariance).

In summary, two factors were extracted from the EFA. The result is consistent with that of past studies (e.g., [12,13]), implying that the conceptual two dimensions (of the FACES-III) also hold for Malaysian adolescents. On the other hand, however, it is important to note that some items (e.g., “children have a say in their discipline” and “we like to do things with just our immediate family”) were excluded due to their low factor loading. In other words, those items are not applicable to our sample, indicating the differences in the meaning of family functioning perceived by Malaysians and Westerners. Moreover, the removal of items also resulted in a new version, 11-item FACES-III. Although the shorter form shows an acceptable reliability, the total explained variance was below 50%. Therefore, further evaluation is needed to understand if the two-factor model with 11 items is superior to the conceptual model and applicable to Malaysian adolescents.

### 3.2. Confirmatory Factor Analysis Results and Discussion

Although our EFA results consistently suggest a two-factor solution, the total explained variance of the 14- and 11-item two-factor models was not satisfactory. Further examination is required to understand if the models are acceptable. On the other hand, a three-factor model has been revealed in past studies. Crowley tested the FACES-III on a mother sample and found an additional factor known as control and discipline (items 2, 4, 10, and 12) [25], besides the cohesion and adaptability factors. Similarly, Ellerman and Strahan found a three-factor model in a sample consisting of children and parents [26]. While the cohesion factor was retained, the adaptability factor was separated into two factors: democracy (items 2, 4, 6, 10, and 12) and change (items 8, 14, 16, 18, and 20). As a result, we examined all the competing models on the validation sample to identify the model that applies to our sample.

A total of 8 models were compared through CFA (see Table 2). The results showed that the one-factor models with 20 items (Model 1), 14 items (Model 2), and 11 items (Model 3) were poor fits. Similarly, the two-factor models with 20 items (Model 4) and 14 items (Model 5) showed a poor fit and unsatisfactory results, respectively. On the other hand, the two-factor model with 11 items (Model 6) demonstrated a good fit. Finally, both three-factor models with 20 items revealed by Crowley [25] (Model 7) and Ellerman and Strahan [26] (Model 8) were also unacceptable.

Taken together, the two-factor model with 11 items that we explored and obtained from EFA is superior to the other models and hence was selected to account for the factorial structure of the FACES-III-M. Pearson’s correlation analysis showed that the two subscales were positively correlated with each other, r(426) = 0.181, *p* < 0.001. Moreover, the (standardized) factor loadings of the 11 items were all statistically significant and were above 0.49 (see Figure 1).

The reliability of the 11-item FACES-III-M was satisfactory. The Cronbach’s alpha for the overall scale was 0.781 (ω = 0.775). The cohesion and adaptability subscales also demonstrated a good to acceptable internal reliability, with values of 0.839 (ω = 0.841) and 0.671 (ω = 0.672), respectively.

In terms of validity, the AVE values of the cohesion and adaptability subscales were 0.433 and 0.339, respectively. Since the AVE < 0.50 in both subscales, the convergent validity of the 11-item FACES-III-M was not supported in the present study. Nevertheless, the 11-item FACES-III-M demonstrated a good discriminant validity, with the square root of the AVE of both subscales being greater than the factor covariance.

## 4. General Discussion

This study examined the psychometric properties of the FACES-III Malay version (FACES-III-M) in a sample of Malaysian adolescents. While the two conceptual factors were replicated, a new and shorter form of the FACES-III-M was found in the present study.

Both the EFA and CFA results supported the hypothetical two-factor structure of the FACES-III proposed by Olson [11]. The findings were also consistent with previous studies (e.g., [12,13]). Although the two dimensions of family functioning were recognized by our sample, a new short form with 11 items (known as the FACES-III-M-SF) outperformed all the competing models. That is, the two-factor model with 11 items (seven items for the cohesion subscale and four items for the adaptability subscale), rather than the original 20 items, was suitable for our sample.

The item removal implies that Malaysian adolescents perceive the meaning of cohesion and adaptability in a different way from Westerners. In terms of cohesion, items such as “we like to do things with just our immediate family” and “family members consult other family members on their decisions” do not seem to adequately reflect the emotional attachment between family members in Malaysia, due to their low factor loading. These acts may represent the usual practice of Malaysian families that practice collectivism [27]. On the other hand, adaptability items such as “in solving problems, the children’s suggestions are followed” and “children have a say in their discipline” reported a poor factor loading. That is, Malaysian adolescents may play a less significant role in determining changes in their family, which may be due to the practice of filial piety [28].

A positive correlation was observed between the cohesion and adaptability factors in both (exploration and validation) samples. Although this finding contradicts Olson’s finding of no correlation between the two factors [11], it is consistent with other studies [15,16,17]. Olson justified that the lack of correlation between cohesion and adaptability represents the ideal Circumplex Model, in which these two dimensions are distinct [11]. However, in reality this may not be the case because, to the best of our knowledge, later studies have consistently failed to replicate this finding even when adopting the FACES-IV [29]. Therefore, cohesion and adaptability are better understood as two dimensions of family functioning that are related to each other, instead of two independent dimensions. 

The FACES-III-M-SF showed satisfactory reliability in both exploration and validation samples. Specifically, the cohesion subscale reported a high reliability, whereas the adaptability subscale showed an acceptable reliability, as in Olson’s study [11]. The results are also consistent with previous studies that demonstrated a higher internal reliability for the cohesion subscale compared to the adaptability subscale (e.g., [12,30]). A possible reason for the low reliability of the adaptability subscale is that the four items are somewhat conceptually different from each other. For instance, items such as “different persons act as leaders in our family” and “it is hard to identify the leader(s) in our family” imply leadership in the family, whereas items such as “the children make the decisions in our family” and “rules change in our family” refer to the family roles and rules. Thus, to improve the reliability of the adaptability subscale, researchers may need to consider refining the items.

The present study is the first to examine the psychometric properties of the Malay version of the FACES-III among Malaysian adolescents. Our findings contribute to the current literature by discovering the best factorial structure of the FACES-III-M (i.e., FACES-III-M-SF with 11 items) that works for Malaysian adolescents. Researchers can then use FACES-III-M-SF to measure adolescents’ family functioning in the context of Malaysia. Moreover, as the conceptual 20-item two-factor model of the FACES-III-M does not fit well with Malaysian adolescents, local researchers are advised to investigate the psychometrics of the other family functioning measurements established in other cultures before administering them to the local community. In line with this, Chin and colleagues also demonstrated that a family functioning developed in a foreign culture (i.e., the Family Assessment Device—General Functioning Subscale) is not appropriate for a Malaysian sample [31].

A limitation of the study is that the validity of the FACES-III-M-SF was not thoroughly examined. In the present study, we only focused on the convergent and discriminant validity. Likewise, even though the FACES-III-M-SF outperformed the other models, the results were derived from the full 20-item version of the FACES-III-M. Furthermore, the FACES-III was not originally developed from an adolescent’s perspective. Thus, future researchers should test the criterion-related validity of the FACES-III-M-SF on a new sample to certify whether the FACES-III-M-SF is applicable for adolescents. Besides that, the convergent validity of the FACES-III-M-SF was neither supported in the EFA nor CFA using the AVE method; therefore, future researchers are advised to further examine the convergent validity by correlating the scores of the 11-item FACES-III-M with another measurement of family functioning. Lastly, our study was only conducted in the state of Kuala Lumpur, which limits the generalizability of the results to the whole of Malaysia. Hence, future researchers should include participants from different states of Malaysia to verify the usability of the FACES-III-M-SF.

## 5. Conclusions

Family functioning is a culturally sensitive concept. Our study found that almost half of the FACES-III items were unable to capture the meaning of family functioning among Malaysian adolescents. Nevertheless, the present study demonstrates that the 11-item FACES-III-M-SF appears to be an adequate measurement of family functioning in the Malaysian context. Future studies are recommended to continue this line of research to refine and improve the FACES-III-M-SF.

## Figures and Tables

**Figure 1 ijerph-19-00156-f001:**
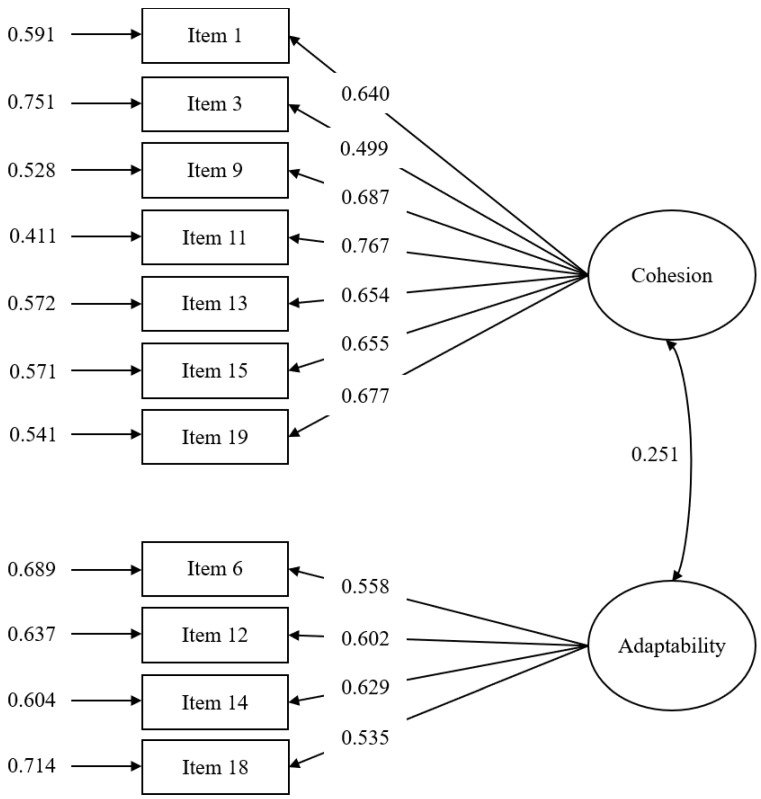
Factorial Structure of the 11-item FACES-III-M (validation sample, *n* = 428).

**Table 1 ijerph-19-00156-t001:** Summary of factor loading by maximum likelihood for the FACES-III-M (exploration sample, *n* = 424).

Items		14-Item: 2-Factor ^a^		11-Item: 2-Factor ^b^	
		F1	F2	F1	F2
1	Ahli keluarga saya saling bantu-membantu di antara satu sama lain (Family members ask each other for help)	0.568	0.077	0.571	0.087
3	Ahli keluarga saya dapat menerima rakan-rakan ahli keluarga yang lain dengan baik (We approve of each other’s friends)	0.519	0.026	0.519	0.034
7	Ahli keluarga berasa lebih rapat dengan ahli keluarga sendiri berbanding orang luar (Family members feel closer to other family members than to people outside the family)	0.455	−0.054	-	-
9	Ahli keluarga saya lebih suka menghabiskan masa bersama dengan keluarga (Family members like to spend free time with each other)	0.613	0.058	0.620	0.028
11	Ahli keluarga saya berasa rapat di antara satu sama lain (Family members feel very close to each other)	0.764	−0.116	0.760	−0.129
13	Apabila keluarga kami berkumpul untuk melakukan aktiviti, semua akan hadir (When our family gets together for activities, everybody is present)	0.633	0.020	0.635	0.023
15	Kami boleh merancang sesuatu dengan mudah untuk dilakukan bersama-sama sebagai satu keluarga (We can easily think of things to do together as a family)	0.711	0.022	0.717	0.027
19	Kebersamaan (togetherness) keluarga adalah sangat penting (Family togetherness is very important)	0.673	−0.065	0.671	−0.079
6	Orang berbeza bertindak sebagai pemimpin dalam keluarga kami (Different persons act as leaders in our family)	0.003	0.504	0.006	0.588
10	Ibu bapa dan anak-anak berbincang tentang bentuk hukuman sesuatu kesalahan secara bersama-sama (Parent(s) and children discuss punishment together)	0.201	0.424	-	-
12	Anak-anak menentukan keputusan dalam keluarga saya (The children make the decisions in our family)	−0.012	0.582	0.008	0.607
14	Peraturan berubah dalam keluarga kami (Rules change in our family)	0.017	0.581	0.059	0.535
18	Adalah sukar untuk mengenalpasti pemimpin dalam keluarga saya (It is hard to identify the leader(s) in our family)	−0.118	0.575	−0.065	0.500
20	Adalah sukar menetukan siapa yang buat apa dalam kerja-kerja rumah (It is hard to tell who does which household chores)	−0.021	0.452	-	-
	Total explained variance (%)	34.5		38.2	
	Cronbach’s alpha coefficient	0.795 ^c^		0.778 ^c^	
		0.825	0.691	0.829	0.641
	McDonald’s omega coefficient	0.784 ^c^		0.772 ^c^	
		0.827	0.692	0.831	0.643

Note. F1 = cohesion, F2 = adaptability, boldface factor loadings are greater than 0.40. ^a^ Items 2, 4, 5, 8, 16, and 17 were removed due to their low factor loading. ^b^ Items 7, 10, and 20 were removed due to their low factor loading. ^c^ Reliability coefficients of the overall scale.

**Table 2 ijerph-19-00156-t002:** Model fit indices from confirmatory factor analysis (validation sample, *n* = 428).

Model		χ^₂^	df	χ^₂^/df	TLI	CFI	RMSEA [90% CI]	SRMR
1	1-Factor (20 items)	730.0 ***	170	4.29	0.704	0.735	0.088 [0.081, 0.094]	0.090
2	1-Factor (14 items)	485.8 ***	77	6.31	0.682	0.731	0.111 [0.102, 0.121]	0.109
3	1-Factor (11 items)	285.6 ***	44	6.49	0.748	0.799	0.113 [0.101, 0.126]	0.101
4	2-Factor (20 items)	517.8 ***	169	3.06	0.814	0.835	0.069 [0.063, 0.076]	0.082
5	2-Factor (14 items)	193.3 ***	76	2.54	0.908	0.923	0.060 [0.050, 0.071]	0.064
6	2-Factor (11 items)	80.2 ***	43	1.87	0.960	0.969	0.045 [0.029, 0.060]	0.043
7	3-Factor (20 items; [25])	484.2 ***	167	2.90	0.829	0.850	0.067 [0.060, 0.074]	0.081
8	3-Factor (20 items; [26])	505.9 ***	167	3.03	0.818	0.840	0.069 [0.062, 0.076]	0.083

Note. TLI = Tucker–Lewis Index, CFI = Comparative Fit Index, RMSEA = Root Mean Square Error of Approximation, CI = confidence interval, SRMR = standardized root mean square residual. *** *p* < 0.001.

## Data Availability

The datasets generated during and/or analyzed in this study are available on request from the first corresponding author.

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
