# Peer review of "Psychometric Evaluation of the Malay Version of the Family Adaptability and Cohesion Evaluation Scale III for Malaysian Adolescents"

_ijerph, 2021, doi:10.3390/ijerph19010156_

Round 1

Reviewer 1 Report

What is the main question addressed by the research?
As I guess, because the goal is not explicitly stated. The aim of the research was to investigate the psychometric properties of the Malay version of FACES-III (FACES-III-M) on the Malaysian sample. Therefore, I recommend that you write directly both in the abstract and in the introduction of the sentence starting with the words: "The purpose of the research was ..." !!! 2. Do you consider the topic original or relevant in the field? Does it address a specific gap in the field?
So the topic is very important, especially for psychometric research in Malaysia. Therefore, the adaptation of the FACES-III-M method to the Malaysian conditions is particularly valuable.
3. What does it add to the subject area compared with other published material?
The topic is very practical and very important, therefore the authors should be especially appreciated for its implementation.
4. What specific improvements should the authors consider regarding the methodology? What further controls should be considered?
The methodology was correctly selected to conduct research in this type of problem. I have no objections.
5. Are the conclusions consistent with the evidence and arguments presented and do they address the main question posed?
The conclusions are correctly formulated and result from the conducted analyzes. The adopted research goal was achieved at a satisfactory level.
6. Are the references appropriate?
The literature cited in the work corresponds to the current research status in the field of the discussed issues.
7. Please include any assitional comments on the tables and figures.
No critical remarks regarding tables and figures

Reviewer 2 Report

     I suggest that the authors follow established standard principles and practices of research writing, especially as it relates to the Results Section. In this manuscript, the Results Section is rampant with the authors’ opinions, comments, deductions, assumptions, inferences, and references which must be reserved for the Discussion Section. The Results Sections MUST be FREE from the authors’ opinions, comments, claims, inferences, and references. It should not be buttressed with material from other studies mingled with the results of this study. Therefore, this reviewer recommends a total rewrite-up of the Results Section, by removing all the authors’ opinions, comments, deductions, assumptions, inferences, and references and replacing these with a descriptive narrative of the results of this study. Accordingly, from the Results Section under Exploratory Factor Analysis, Pages 5, lines 166-176, the following paragraph, should be removed, and if necessary, moved to the Discussion Section: 

In sum, two factors were extracted from the EFA. The result is consistent with the past studies (e.g., Forjaz et al., 2002; Hasui et al., 2004) implying that the conceptual two dimensions (of the FACES-III) also hold for Malaysian adolescents. On the other hand, however, it is important to note that some items (e.g., “children have a say in their discipline”, and “we like to do things with just our immediate family”) were excluded due to low factor loading. In other words, those items are not applicable to our sample, indicating the differences in the meaning of family functioning perceived by Malaysians and Westerners. Moreover, the removal of items also resulted in a new version, 11-item FACES-III. Although the shorter form shows acceptable reliability, the total explained variance was below 50%. Therefore, further evaluation is needed to understand if the two-factor model with 11 items is superior to the conceptual model and applicable to Malaysian adolescents.

 Moreover, the Results Section under Confirmatory Factor Analysis, Page 5, lines 178-196 needs a total rewrite-up, by removing all the authors’ opinions, comments, deductions, assumptions, inferences, and references (all of which should belong to the Discussion Section) and replacing these with a descriptive narrative of the facts only from results of this study.

Round 2

Reviewer 2 Report

This is NOT a standard structure of a manuscript. The papers needs to follow the IMRAD(Introduction, Method, Results, and Discussion) format  or one of the many standard manuscript writing formats. If author guidelines permit  Results and Discussion to be under the same subtitle/section, that is fine but then the paper cannot have another separate section for the Discussion. A stand alone Results Section should be free from inferences, references, and authors opinions and comments in its narrative.   In sum, this brief report is flawed in design, amorphous in  structure, and does not follow proper procedures for the adaptation of a research instrument into a different language, and the newly adapted instrument has not been piloted on a sample population to determine it validity and reliability . (Please refer to my review for the original submission of this manuscript. )

Reference  

https://f1000research.com/for-authors/article-guidelines/brief-report
